# Biochemical, Clinical, and Genetic Characteristics of Mexican Patients with Primary Hypertriglyceridemia, Including the First Case of Hyperchylomicronemia Syndrome Due to GPIHBP1 Deficiency

**DOI:** 10.3390/ijms24010465

**Published:** 2022-12-27

**Authors:** Perla Graciela Rodríguez-Gutiérrez, Ana Gabriela Colima-Fausto, Paola Montserrat Zepeda-Olmos, Teresita de Jesús Hernández-Flores, Juan Ramón González-García, María Teresa Magaña-Torres

**Affiliations:** 1División de Genética, Centro de Investigación Biomédica de Occidente, Instituto Mexicano del Seguro Social, Sierra Mojada 800, Independencia Oriente, Guadalajara 44340, Jalisco, Mexico; 2Doctorado en Genética Humana, Centro Universitario de Ciencias de la Salud, Universidad de Guadalajara, Sierra Mojada 950, Independencia Oriente, Guadalajara 44340, Jalisco, Mexico; 3School of Medicine, Universidad Autónoma de Guadalajara, Universidad 700, Lomas del Valle, Guadalajara 45129, Jalisco, Mexico; 4Departamento de Disciplinas Filosóficas Metodológicas e Instrumentales, Centro Universitario de Ciencias de la Salud, Universidad de Guadalajara, Sierra Mojada 950, Independencia Oriente, Guadalajara 44340, Jalisco, Mexico; 5Unidad de VIH, Hospital Civil de Guadalajara “Fray Antonio Alcalde”, Guadalajara 44280, Jalisco, Mexico

**Keywords:** primary hypertriglyceridemia, hyperchylomicronemia, *APOA5*, *GPIHBP1*, *LMF1*, *LPL*, Mexicans

## Abstract

Primary hypertriglyceridemia (PHTG) is characterized by a high concentration of triglycerides (TG); it is divided between familial hyperchylomicronemia syndrome and multifactorial chylomicronemia syndrome. In Mexico, hypertriglyceridemia constitutes a health problem in which the genetic bases have been scarcely explored; therefore, our objective was to describe biochemical–clinical characteristics and variants in the *APOA5, GPIHBP1, LMF1,* and *LPL* genes in patients with primary hypertriglyceridemia. Thirty DNA fragments were analyzed using PCR and Sanger sequencing in 58 unrelated patients. The patients’ main clinical–biochemical features were hypoalphalipoproteinemia (77.6%), pancreatitis (18.1%), and a TG median value of 773.9 mg/dL. A total of 74 variants were found (10 in *APOA5*, 16 in *GPIHBP1*, 34 in *LMF1*, and 14 in *LPL*), of which 15 could be involved in the development of PHTG: 3 common variants with significative odds and 12 heterozygous rare pathogenic variants distributed in 12 patients. We report on the first Mexican patient with hyperchylomicronemia syndrome due to GPIHBP1 deficiency caused by three variants: p.R145*, p.A154_G155insK, and p.A154Rfs*152. Moreover, eleven patients were heterozygous for the rare variants described as causing PHTG and also presented common variants of risk, which could partially explain their phenotype. In terms of findings, two novel genetic variants, c.-40_-22del *LMF1* and p.G242Dfs*10 *LPL,* were identified.

## 1. Introduction

Primary hypertriglyceridemia (PHTG) is a genetic disorder characterized by a high concentration of triglycerides (TG) in plasma (175–885 mg/dL) and is a risk factor of pancreatitis, cardiovascular diseases, stroke, and high blood pressure. In general, two types of PHTG have been described: familial hyperchylomicronemia syndrome (FCS) and multifactorial chylomicronemia syndrome (MCS). FCS is a rare autosomal recessive disease with a prevalence of 1 in 1,000,000 and is clinically distinguished by xanthomas and lipemia retinalis. Biochemically, it presents a severe hypertriglyceridemia (HTG) (TG ≥ 885 mg/dL) as a consequence of chylomicron accumulation and a scarcity of VLDL, chylomicron remnants, and other lipoproteins, all of which result from a lipolytic blockade that compromises the conversion of large TG-carrying particles into smaller lipoproteins [1]. Genetic causes of PHTG are usually pathogenic variants (homozygous or compound heterozygous) in the *LPL, APOC2, APOA5, LMF1,* and *GPIHBP1* genes, rendering haploinsufficiency or loss of function. MCS has an unclear inheritance pattern and a prevalence estimated of ~1 in 600. In patients with MCS, whose lipolysis pathway is partially blocked, VLDL and remnants are present in abundance and apoB-100 levels are slightly high. The rising of TG in these patients is due to both rare and common variants in the five genes mentioned above, as well as to the influence of environmental factors [1].

Although more than 50 genes are involved in TG metabolism, less than 20% have been related to the development of PHTG. About 90% of FCS patients present pathogenic variants in the *LPL* gene and less than 10% in the *APOC2, APOA5, LMF1,* and *GPIHBP1* genes [2]. Inasmuch as LPL function is enhanced by the products of these genes, their allelic variants will directly render an insufficient activity of LPL and the consequent increase in TG [1].

LPL is one of the most important proteins in the metabolism of TG. It forms dimers in order to become activated and, through GPIHBP1 and heparan sulfate proteoglycans, binds to capillary endothelial cells, where it performs its main functions: to hydrolyze TG from triglyceride-rich lipoproteins and to act as a ligand, facilitating the uptake of lipoproteins by receptors, such as LDLR and LRP1 [3]. On the other hand, LMF1 is involved in the maturation processes of LPL, hepatic lipase, and endothelial lipase [4]. Although APOA5′s function is unknown, at least two hypotheses have been proposed: (1) the catabolic enhancing of TG-rich lipoproteins through activation of LPL; and, (2) the inhibition of VLDL production, which is the main carrier of TG [5]. A significant number of variants in these four genes have been related to PHTG: 19 in *APOA5*, 60 in *GPIHBP1*, 50 in *LMF1*, and ~300 in *LPL* [6].

Despite HTG’s being one of the most frequent dyslipidemias and an important health problem in Mexico (17.3%) [7], an insignificant number of patients have been studied, rendering information on the genetic basis of the disease scarce. Since only three causal variants have been described in Mexican patients so far: two in *LPL* (p.G188E and c.94_98del) and the other in *LMF1* (p.Y439*) [8,9,10], this study aimed to analyze clinical–biochemical characteristics and variants in the *APOA5*, *GPIHBP1*, *LMF1*, and *LPL* genes of patients with primary hypertriglyceridemia. As a result, we found two novel variants and the first Mexican patient with hyperchylomicronemia syndrome due to GPIHBP1 deficiency.

## 2. Results and Discussion

### 2.1. Description of Biochemical and Clinical Characteristics of Patients with Primary Hypertriglyceridemia

Even though individuals with high levels of TG are common in Mexico (17.3%) [7], there is a scarcity of specific studies describing clinical, biochemical, and genetic characteristics of PHTG [8,9,10]. Table 1 and Figure 1 summarize the biochemical and clinical characteristics of the 58 index cases with diagnosis of PHTG, who were classified in three groups: FCS (*n* = 4), MCS children (*n* = 7), and MCS adults (*n* = 47). The comparison of quantitative variables among the three groups showed significant differences for the two most relevant lipids in PHTG: TG were higher in FCS patients (2826 mg/dL) than in either MCS children (475.1 mg/dL, *p* < 0.01) or MCS adults (791 mg/dL, *p* < 0.01); by contrast, HDL levels were lower in FCS patients (11.1 mg/dL) than in either MCS children (32.8 mg/dL, *p* = 0.02) or MCS adults (30.5 mg/dL, *p* < 0.01). Remarkably, the HDL mean value for each group was lower than 35 mg/dL, which is consistent with the high percentage of patients with hypoalphalipoproteinemia (77.6%).

Significant differences for several variables stood out when our results were compared with those observed in other populations. The mean age of our patients (37.5 ± 17.5 years, *n* = 58) was significantly lower than that reported in French patients (43.2 ± 16.2 years, *n* = 385; *p* = 0.01) [11], but similar to that observed in Chinese patients (37.38 ± 11.02 years, *n* = 40; *p* = 0.99) [12]. TG median value in our patients (773.9 [205–9848] mg/dL) was not significantly different from those reported either in Chinese (TG 850.3 ± 345.4 mg/dL, *p* = 0.06) [13] or in European patients (TG 1151.4 ± 1479.1 mg/dL, *p* = 0.77) [14]. On the other hand, our patients with clinical and biochemical diagnosis of FCS had severe HTG and their TG levels were similar to those observed by Hegele et al. in patients with variants in the *LPL* gene (TG 2816 ± 839.1 mg/dL vs. 2613 ± 1307 mg/dL, *p* = 0.36) and in other genes (2816 ± 839.1 mg/dL vs. 1849 ± 938, *p* = 0.09) [2].

Hypoalphalipoproteinemia is a common dyslipidemia in patients with PHTG and represents a risk factor for cardiovascular diseases. In this study, 77.6% (45/58) of patients had hypoalphalipoproteinemia; similar percentages had been observed in populations from Italy (72.7%, 8/11; *p* = 0.73) [15] and Korea (73.1%, 19/26; *p* = 0.65) [16], while Japanese patients were significantly different, as their percentages were lower (45.5%, 10/22; *p* = 0.006) [17]. An individual’s lipid levels are the consequence of both modifiable (overweight, high carbohydrate intake, smoking, and the lacking of regular physical activity, among others) and non-modifiable factors (variants in genes, such as *ABCA1*, *APOA1,* and *LCAT* [18]), which may explain the variability of these results among populations.

Biochemical, clinical, and molecular characteristics of all 58 patients are detailed in Appendix A. Both HTG-specific and concomitant diseases were detected: hypertriglyceridemia-induced acute pancreatitis (HTG-IAP), 18.1%; xanthomas, 0%; lipemia retinalis, 0%; arterial hypertension, 35.1% (adults); type 2 diabetes mellitus, 31.1% (adults); hypothyroidism, 10.2%; and cardiovascular disease, 4.5% (adults). Pancreatitis is one of the most severe clinical features that these patients present; four of our patients had multiple episodes of pancreatitis (from four to nine events) and six had at least one. The percentage of this complication was similar to that reported in populations from China (17.5%, 18/103, *p* = 0.95) [13], Korea (3.8%, 1/26, *p* = 0.08) [16], and Japan (4.3%, 1/23, *p* = 0.12) [17].

### 2.2. DNA Analysis

The molecular analysis of the 58 patients disclosed 74 variants: 10 in *APOA5*, 16 in *GPIHBP1*, 34 in *LMF1*, and 14 in *LPL* (Table 2 and Table 3). A total of 3 variants were located in promoter regions, 42 were in exons (20 missense, 16 synonym, 3 nonsense, 2 deletions, and 1 insertion), 25 in introns, and 4 in 3′ untranslated regions. A great genetic heterogeneity was observed in the studied group: 47 variants (63.5%) were rare (mutated allele frequency <5%) and 27 (36.5%) were common (mutated allele frequency >5%); the 2 most frequent variants were c.295 + 27T > C *GPIHBP1* (76.7%) and c.162-43 A > G *APOA5* (74.1%). Overall, each patient had a minimum of 8 and a maximum of 22 variants of the four screened genes. To our knowledge, there are no studies regarding 49 of these variants (16 common and 33 rare) that would allow us to understand their role in the development of HTG. Moreover, 34 (72.3%) of the rare variants have not been identified in the Mexican population.

Two variants, c.-40_-22del *LMF1* and p.G242Dfs*10 *LPL,* have not been previously reported in the consulted databases. We performed in silico analyses to determine their potential effect on proteins. The p.G242Dfs*10 *LPL* variant was classified as disease-causing with the *Mutation Taster* program and will be described in detail later. For the c.-40_-22del *LMF1* deletion, with the *PROMO* program, we detected the loss of a motif sequence necessary for the binding of nine different transcription factors: GRα, AP-2alpha, GRβ, PAX5, TP53, E2F-1, Sp1, and YY1. The c.-41_22del variant is a 19 bp deletion within the promoter of the *LMF1* gene that had a frequency of 36.2% in our population. Even though in silico analyses revealed that the binding sites for nine transcription factors could be modified with this deletion, the screening of the index cases’ relatives showed that individuals homozygous for the variant were normotriglyceridemic, which would disqualify it as causal of dyslipidemia. Although the promoter sequence of the *LMF1* gene is highly repetitive and a large number of deletions and insertions have been described [19], functional studies are needed to know its effects on transcription.

### 2.3. Odds Ratio Analyses in Common Genetic Variants

In order to know if the common variants detected in patients could represent a risk factor for developing PHTG, we compared the genotype frequencies of 27 variants with information available in the 1000 Genomes Project database [19]. Three variants showed significant differences:

The rs651821 (c.-3A > G) *APOA5*: we found that 58.6% of patients carried the risk allele G (AG or GG genotypes), which, under the dominant model (AA vs. AG + GG), had an OR value of 2.9 (95% CI 1.39–6.1, *p* = 0.005). The G allele has been associated with a high risk of developing dyslipidemia and coronary artery disease (CAD), as well as with increased TG [20,21,22]. Jorgensen et al. determined that AG and GG genotypes promoted a TG increase of 18 and 51%, respectively [22]. Chou et al. found that the transcription factor GATA 4 has higher affinity for binding to the GATA sequence than to the GGTA sequence, which is why the G allele decreases the transcription rate of the *APOA5* gene [23].

The rs3135506 (c.56 C > G, p.S19W) *APOA5*: 50% of our patients had the risk allele G, which, under the dominant model (CC vs. CG + GG), displayed a significant OR = 3.9 (95% CI 1.77–8.71, *p* = 0.0008). CG and GG genotypes have been linked to a high risk of developing chronic diseases, such as high blood pressure [24], obesity [25], CAD [26], and acute myocardial infarction [27]. On the other hand, the G allele, in different populations, has been associated with high levels of TG, and studies demonstrated that this variant reduces APOA5 activity up to 49% by modifying the insertion angle of the protein in the lipid membrane from 40° to 65° [28], which could explain its relation with some diseases and variables. It should be noted that variants rs3135506 and rs12287066 (c.132 C>A, p.I44I) were in linkage disequilibrium, and no associations with the last variant, which is synonymous, have been reported.

The rs3751666 (c.194-28T > C) *LMF1*: in this study, under the recessive model (TT + CT vs. CC), the CC genotype turned out to be a risk factor (OR = 4.0, 95% CI 1.7–9.7, *p* = 0.002) and 39.6% of our patients were homozygous CC. Although this common variant has not been widely studied, Hosseini et al. found the CC genotype to be associated with lower LPL activity than TT and CT genotypes. These authors suggested that the rs3751666 variant could be in linkage disequilibrium with other functional common variants affecting *LMF1* expression or function, although they did not rule out the possibility that this variant located at 28 bp from exon 2 could directly affect the LMF1 RNA splicing. They also determined that the variation in *LMF1* expression was associated with changes in LPL activities [29].

### 2.4. Description of the Common and Rare Genetic Variants Related to Primary Hypertriglyceridemia, as Well as the Clinical Characteristics of the Patients

Considering our analyses and the information reported in the databases [6,19,30], we can suggest that at least 15 variants detected in this study may be influencing the PHTG phenotype in Mexican patients. Of these 15, 3 were common variants with significant odds ratio indicative of risk factor (rs651821, rs3135506, and rs3751666, described above), whereas the other 12 were heterozygous rare variants, distributed among 12 (20.7%) patients (one of them with FCS and 11 with MCS—heterozygous), which had been classified as pathogenic: 2 in *APOA5* (p.W120* and p.G185C), 4 in *GPIHBP1* (c.53-2A > G, p.R145*, p.A154Rfs*152, and p.A154_G155 insK), 1 in *LMF1* 8 (p.A469T), and 5 in *LPL* (p.H71Q, p.G215E, p.G242Dfs*10, p.N318S, and p.K410N). Except for variants p.N318S *LPL* (*n* = 2) and p.G185C *APOAV* (*n* = 2), all were found in a single patient. Even though the largest number of variants was detected in the *LMF1* gene, 34 (45.9%), we were unable to relate 32 of them with dyslipidemia, mainly due to a lack of information. In the following sentences, we will describe the relevant characteristics of the aforementioned 15 variants and the patients who carried them.

Case report p.R145*, p.A154Rfs*152, and p.A154_G155insK *GPIHBP1*: these variants were identified in an adult male patient referred to a genetic consultation for persistent severe HTG (TG 3300 mg/dL) since childhood and with a history of poor response to treatment with bezafibrate; unfortunately, the complementary clinical features were not provided. He represents a compound heterozygous case with three variants in the *GPIHBP1* gene: p.R145*, p.A154Rfs*152, and p.A154_G155 insK (patient No. 46, Appendix A).

The p.R145* causes loss of the last 40 amino acids of GPIHBP1, thus producing a protein of 144 amino acids; and p.A154Rfs*152 is the consequence of 460-461 GC deletion, generating a longer protein (from 184 to 304 amino acids) with a reading frame shift from codon 154. The p.A154_G155insK results from an adenine triplet insertion at codon 462, which translates to a lysine at residue 155.

The p.R145* and p.A154Rfs*152 variants modify an important segment of the GPIHBP1 propeptide domain (152–184 amino acids), which, in the endoplasmic reticulum, is normally removed and replaced by the glycosylphosphatidylinositol (GPI) anchor, whose function is to bind GPIHBP1 protein onto the cell surface. A functional hydrophobic amino acid sequence (164–179 aa) is located within this propeptide domain, which triggers the addition of the GPI anchor [31]. Moreover, lysine insertion at 155 amino acid position is near a hydrophobic amino acid domain and could change the affinity of the protein complex involved in the incorporation of the GPI anchor.

The variants p.A154Rfs*152 and p.A154_G155insK were in *cis*, leading a 305 aa, instead of the expected 184 aa protein (Figure 2). Noteworthily, p.R145* and p.A154Rfs*152 abolished the carboxyl-terminal domain function, so we suggest that the patient had a deficiency of the GPIHBP1 protein, which caused FCS. To our knowledge, two Mexican patients with FCS have been reported, both were LPL deficient [8,10]. Although this patient constitutes the third case, he is the first with variants in the *GPIHPB1* gene.

The remaining nine pathogenic variants were distributed among 11 patients with MCS-compatible phenotypes. These patients presented heterogeneous clinical features and, in addition to the pathogenic variant, they showed other genetic variants, which have been scarcely studied (Appendix A).

The p.W120* *APOA5*: this variant was detected in a 28-year-old male patient with moderate HTG (TG 688 mg/dL) and poor response to bezafibrate; he was also heterozygous for rs651821AG and rs3751666TC variants. The p.W120* produces a protein that lacks 247 amino acids, which represent 67.4% of its totality; it has only been reported in the VarSome database and classified as pathogenic by four in silico programs (Mutation Taster, Fathmm-MKL, EIGEN, and Bayes Del) [30].

The p.G185C *APOA5*: this variant was identified in two patients, the first was a 12-year-old overweight girl with high TG (>400 mg/dL) who was also carrier of two common variants, rs3135506CG and rs651821AG. She reduced her caloric intake and increased exercise activities to normalize her weight, but her TG levels remained moderately high (358 mg/dL); however, when she was treated with bezafibrate, her TG dropped to 203 mg/dL. The second patient was a 44-year-old man who learned of his HTG (TG 800–1800 mg/dL) when he was 21 years old; he showed to be heterozygous with p.G185C, rs651821AG, and rs3751666TC. His father, at the age of 49, had a myocardial infarction whose apparent main factor was a high TG level. After this event, the whole family underwent a lipid profile where both our index case and one sister revealed severely high TG levels (>800 mg/dL). Clinically, our patient presented dizziness and weakness, which disappeared after starting a daily treatment with 900 mg of gemfibrozil; additionally, TG were significantly reduced (180 mg/dL).

The p.G185C variant that introduces a cysteine at residue 185 allows the protein to form disulfide bonds with various plasma proteins, impairing APOA5 binding to lipoproteins (such as VLDL and HDL) and abrogating its ability to modulate plasma TG levels [32]. The C allele has been identified as a risk factor for HTG (OR = 3.6, 95% CI: 2.5–5.1, *p* = 0.000) and has also been associated with increased levels of TG [33]. Yang et al. reported a patient with mild HTG and acute pancreatitis caused by the interaction of two heterozygous variants, p.G185C *APOA5* and p.I252M *LPL* (digenic inheritance), with a history of high alcohol consumption as an environmental factor [34].

The c.53-2A > G *GPIHBP1*: this variant was found together with two other common variants (rs651821AG and rs3751666TC) in a 20-year-old hypertensive man with moderate HTG. The c.53-2A > G variant has been previously reported in only one heterozygous subject [35] and, according to the VarSome database, it is harmful because it may cause the loss of the splicing acceptor site, producing the deletion of exon 2 (129 nt) and generating a protein of 141 aa, with a loss of 43 aa (p.G18_R60del).

The p.A469T *LMF1*: we identified this variant in a patient with severe HTG (TG ranging from 1000 to 1500 mg/dL) who was diagnosed at the age of 35 and who, three years later, developed hypertension. In addition to being heterozygous for the p.A469T variant, he was carrier of rs651821AG. The p.A469T variant, placed in the C-terminal domain, has been described in French (*n* = 1/385) [11] and in Mexican patients with HTG (*n* = 3/728) [9]. Although an in vitro study revealed that the variant does not modify LPL activity [11], 13 in silico predictions cited in VarSome database classified it as damaging [30]; in addition, conflicting interpretations of pathogenicity are reported in ClinVar database [6].

The p.H71Q *LPL*: the patient carrying this variant was also heterozygous for rs3135506CG. At two months of age, he presented severe HTG (TG 3514 mg/dL) and other clinical characteristics independent of dyslipidemia (ostium secundum and pulmonary hypertension). The p.H71Q variant affects the amino-terminal region that contains both a lipase fold initiation site and an interaction site with protein APOC2 (LPL activation cofactor) [36]. Despite the fact that the functional status of p.H71Q has been reported with conflicting interpretations of pathogenicity (ClinVar), Minicocci et al. found it associated with familial combined hyperlipidemia (OR = 16.0, 95% CI 2.0–128.5, *p* = 0.0005) [37].

The p.G215E *LPL*: in this study, a 46-year-old male patient with severe HTG (TG 3859 mg/dL), four episodes of HTG-IAP, and no history of smoking or alcoholism, was heterozygous for the p.G215E, rs3135506CG, and rs3751666TC variants. The variant p.G215E produces unstable dimers that readily dissociate into inactive monomers, rendering the LPL protein unable to perform its function [38]. Moreover, 23.5% of the patients with LPL deficiency have this variant, which has been linked to CAD by both increasing TG levels up to 80% and lowering HDL [39].

The p.G242Dfs*10 *LPL* is a novel variant, which resulted from an adenine deletion at nucleotide 723. This deletion produces a truncated protein of 250 amino acids that lacks 225 residues, resulting in the loss both of a segment of the amino-terminal domain and of the entire carboxyl-terminal domain. These regions are essential for the binding of the LPL protein to lipid and heparin molecules, as well as to the GPIHBP1 and LRP1 proteins [36,38,40,41]. This variant was found in a 37-year-old woman who was also homozygous for the rs3751666 CC common-risk variant; she presented her first episode of HTG-IAP at age 35, after a moderate intake of alcohol. Following this first event, six other episodes took place over the two following years, her TG ranging from 3550 to 9848 mg/d. Factors, such as diet, exercise, and drug intake (ezetimibe 10mg and bezafibrate 200 mg), contributed to a decrease in TG to less than 500 mg/dL.

The p.N318S *LPL*: two heterozygous patients were identified: (1) The first is a 43-year-old woman who suffered pancreatitis at age 32 and who subsequently developed type 2 diabetes mellitus; even with pharmacological treatment (bezafibrate, ezetimibe, and atorvastatin) and diet, her TG levels were in the range of 700–3500 mg/dL. She was heterozygous for p.N318S and rs3135506CG, as well as homozygous for rs651821GG and rs3751666CC. (2) The second patient is a 40-year-old woman whose dyslipidemia was evidenced at age 30 due to a pancreatitis episode, which was recurrent on six subsequent occasions despite having been treated with bezafibrate, ezetimibe, and atorvastatin. The documented TG level was 1332 mg/dL with no antecedents of alcohol drinking, suggesting that, in addition to high TG, she must have had other risk factors for recurrent pancreatitis. The patient was heterozygous for the p.N318S, rs651821AG, and rs3751666TC variants.

The p.N318S variant is in the carboxyl-terminal domain and studies have demonstrated that the p.318S allele causes a decrease in LPL activity of up to 60% due to an increased rate of dissociation from active dimers to inactive monomers; in addition, this variant has been associated with elevated TG and low HDL-C levels [38].

The p.K410N *LPL*: we detected a heterozygous patient for the variant who was also carrier of three common-risk variants: rs651821AG, rs313550 CG, and rs3751666CC. She is a 49-year-old woman without episodes of pancreatitis, but with persistent moderate HTG despite pharmacological treatment (bezafibrate). This variant is located within the PLAT domain (residues 341 to 464), which may be involved in protein–protein and protein–lipid interactions [19]; to our knowledge, there are no in vitro or in vivo studies to determine its functional effect, although the SIFT and MCAP software revealed that it is deleterious, as cited by VarSome database [30]. Association studies have not been performed to determine its implications over any disease.

## 3. Materials and Methods

### 3.1. Subjects

Lipid profile, blood chemistry, and clinical history were performed for 1560 individuals belonging to 250 families, where at least one individual had documented recurrent HTG. Since fifty-eight of those families had biochemical features compatible with PHTG, we selected 58 index cases. To define the FCS and MCS phenotypes, some criteria proposed by Moullin et al. [42] were considered. For FCS: very high TG levels (>885 mg/dL) from the first decade of life and without secondary factors; a TG/TC ratio >5; epigastric pain; hypertriglyceridemia-induced acute pancreatitis; and homozygous or composite heterozygous coding variant *APOA5, GPIHBP1, LPL,* or *LMF1*. For MCS: moderate TG levels (>200 mg/dL), secondary factors triggering HTG (high alcohol intake, features of metabolic syndrome, overweight), TG reduction with treatment and diet, and a combination of a heterozygous loss of function and/or likely pathogenic frequent variants in four genes involved in TG metabolism. In addition to these criteria, we quantified TG in first-degree relatives of the index case and also observed at least one affected individual (some families were biochemically tested multiple times to define the trait).

For biochemical quantifications, a blood sample was collected from each individual with at least 9 h of fasting and without alcohol consumption in the previous 72 h. All studies were carried out using enzymatic methods with commercial kits. In particular, the 58 index cases diagnosed with PHTG gave their informed consent for a second blood draw, which was used for DNA extraction and subsequent molecular analysis. The protocol was structured according to the Declaration of Helsinki guidelines, and it was approved by both the Research and Ethics Committees at our Western Biomedical Research Center-IMSS.

### 3.2. DNA Analysis

After obtaining the DNA by conventional methods, coding and adjacent intronic sequences, as well as promoter and UTR regions of the *APOA5, GPIHBP1, LMF1*, and *LPL* genes, were amplified using polymerase chain reaction (PCR) and sequenced with Big Dye Terminator v3.1 kit. The 30 primers were designed with Oligo 6 software and their sequences are shown in Appendix A. PCR conditions were diverse and are available upon email request. Sequencing reaction was made in a volume of 10 µl containing: 200–500 ηg DNA, 2.5 ρmol of primer, 1.5 µL of 5X buffer, and 0.3 µL of Ready Reaction Big Dye Terminator kit v.3.1. For the thirty sequencing reactions the following program was used: 96 °C 4 min, 25 cycles of 96 °C for 10 seg, 55 °C for 5 seg, and 60 °C for 2 min; finally, 60 °C for 5 min. Sequencing products were purified with columns containing sephadex G-50. Capillary electrophoresis was carried out on an ABI PRISM 310 sequencer and electropherograms were analyzed both manually and with software SnackVar [43]. The DNA and protein reference sequences used to identify variants were NM_052968.5/NP_443200.2 (*APOA5*), NM_178172.6/NP:835466.2 (*GP1HBP1*), NM_022773.4/NP_073610.2 (*LMF1*), and NM_000237.3/NP_000228.1 (*LPL*). Variants were named following the nomenclature recommended by the Human Genome Variation Society [44]. To know whether a specific variant had been previously reported, four databases were consulted: ClinVar [6], Ensembl [19], PubMed, and VarSome [30]. Bioinformatic tools, PROMO [45] and Mutation Taster [46], were used to establish a possible functional effect of the undescribed variants.

### 3.3. Statistical Analysis

Biochemical variables were compared using parametric (student’s *t*-test) or non-parametric (U-Mann–Whitney test) statistics based on data distributions, performing lipid levels adjustment in patients under lipid-lowering treatment [47,48,49,50,51,52]. Clinical characteristics were compared with the chi-square test. Genotype and allele variant frequencies were obtained by direct counting. Odds ratio (OR) analyses were achieved for common variants (allele frequency above 5%) using as reference that reported in the 1000 genomes project database (general population with Mexican ancestry (MXL) and residents in Los Angeles, CA, USA) [19].

## 4. Conclusions

By screening the *APOA5, GPIHBP1, LMF1,* and *LPL* genes in 58 patients with PHTG, 74 variants were found. Twelve pathogenic variants were identified, all of them in the heterozygous state and distributed in 12 patients. Given that one of these patients was compound heterozygous, we are reporting the first Mexican patient with hyperchylomicronemia syndrome caused by three variants in the *GPIHBP1* gene (p.R145*, p.A154Rfs*152, and p.A154_G155insK).

Three common variants were related to the development of PHTG in Mexican patients: rs651821, rs3135506, and rs3751666.

Furthermore, two novel variants, c.-40_-22del *LMF1* and p.G242Dfs*10 *LPL,* were found; the first had high allelic frequency (36.2%) and the second was detected in a single heterozygous patient (0.9%).

As previously described, multifactorial chylomicronemia is the most common multifactorial, polygenic dyslipidemia with complex features resulting from the interactions of genetic and environmental factors and whose etiological bases have been difficult to elucidate. Since the minimum number of variants sufficient to trigger HTG remains unknown and the pathogenicity of many of them is still uncertain, it is clear that in vivo and in vitro studies, as well as family segregation and association analyses, are necessary to increase the information on the causal variants of this frequent, degenerative, and chronic disease.

Finally, one of the limitations of this study is that we were not able to detect variants as large deletions or duplications due to the methodology used (sanger-type sequencing).

## Figures and Tables

**Figure 1 ijms-24-00465-f001:**
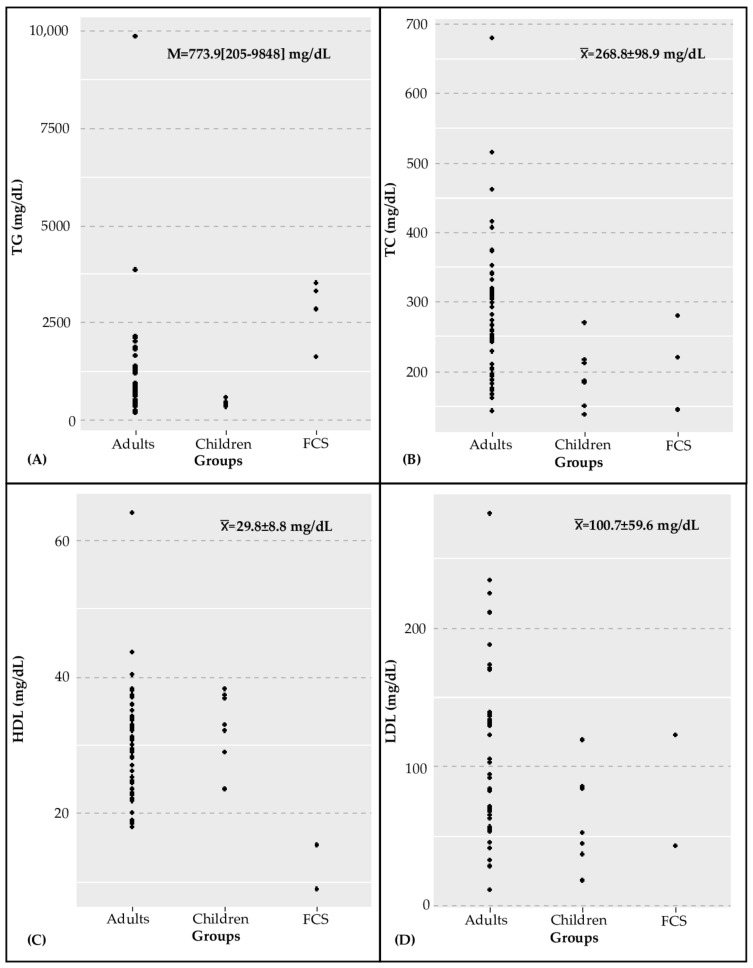
Scatter plot of lipid profile of 58 patients with hypertriglyceridemia. (**a**) Triglyceride (TG); median and range values. (**b**) Total cholesterol (TC); (**c**) High-density lipoprotein; and (**d**) Low-density lipoprotein; mean and standard deviations of these last three variables are given.

**Figure 2 ijms-24-00465-f002:**
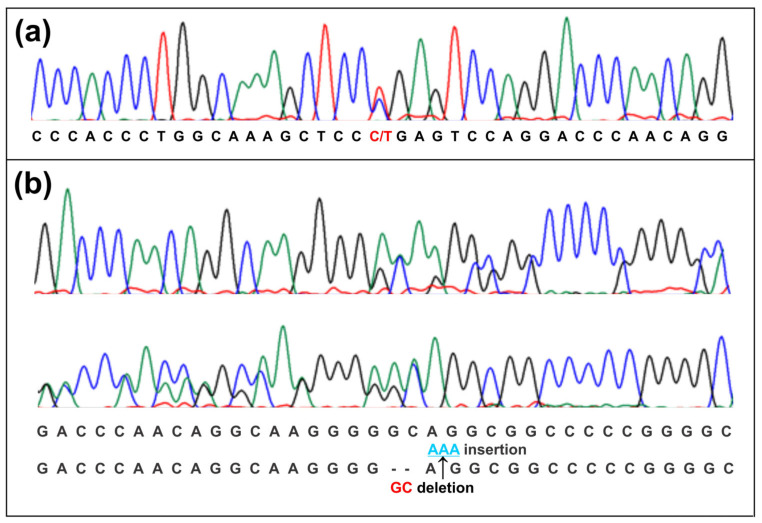
Electropherograms from patient with familial hyperchylomicronemia syndrome showing the sequencing of exon 4 of the *GPIHBP1* gene. (**a**) Forward sequenced region showing a transition C > T in heterozygous state, leading to the p.R145* variant; (**b**) forward and reverse sequenced region displaying both a GC deletion and an AAA insertion in *cis*, rendering the changes p.A154Rfs*152 and p.A154_G155insK, respectively.

**Table 1 ijms-24-00465-t001:** Biochemical and clinical characteristics of patients with primary hypertriglyceridemia.

Variable	Reference Values	Total *n* = 58	FCS *n* = 4	MCS Children *n* = 7	MCS Adults *n* = 47		Values	
						*p* ^c^	*p* ^d^	*p* ^e^
Sex (F/M)		35/23	2/2	4/3	29/18			
Age (y)		37.4 ± 17.7	5.4 ± 4.6	9.4 ± 4.3	44.0 ± 11.7			
Biochemical characteristics								
Glucose ^a^	70–100 mg/dL	96.9 ± 47.8	66.3 ± 10.8	77.8 ± 13.7	102 ± 51.2	0.38	0.04	0.16
Urea ^b^	15–45 mg/dL	25.8 ± 8.8	17.6 ± 4.7	21.1 ± 8.2	27.1 ± 8.7	0.52	0.07	0.12
Creatinine ^a^	0.52–1.2 mg/dL	0.7 [0.1–17.7] ^f^	0.5 ± 0.1	0.5 ± 0.2	0.7 [0.4–17.7] ^f^	0.26	0.01	0.29
TC ^a^	<200 mg/dL	268.8 ± 98.9	215.1 ± 67.6	193.3 ± 44.2	283.8 ± 101	0.52	0.23	<0.01
HDL ^a^	>35 mg/dL	29.8 ± 8.8	11.1 ± 3.7	32.8 ± 5.3	30.5 ± 8.1	0.02	<0.01	0.28
LDL ^a^	<130 mg/dL	100.7 ± 59.6	82.7 ± 56.2	63 ± 34.7	108 ± 61.4	0.67	0.50	0.04
TG ^a^	<200 mg/dL	773.9 [205–9848] ^f^	2826 ± 839.1	475.1 ± 88.4	791 [205–9848] ^f^	<0.01	<0.01	0.01
SBP ^b^	<140 mmHg				124.5 ± 17.5			
DBP ^a^	<90 mmHg				84.3 ± 14.9			
Clinical characteristics								
Arterial hypertension		35.1% (13/37)	0%	0%	35.1% (13/37)			
Cardiovascular disease		3.6% (2/56)	0%	0%	4.5% (2/45)			
Hypoalphalipoproteinemia		77.6% (45/58)	100% (4/4)	57.1% (4/7)	78.7% (37(47)			
Hypothyroidism		10.2% (5/49)	0%	14.3% (1/7)	9.5% (4/42)			
Pancreatitis		18.1% (10/55)	50% (2/4)	0%	17.8% (8/45)			
Type 2 diabetes mellitus		25.5% (14/55)	0%	0%	31.1% (14/45)			

DBP: diastolic blood pressure, FCS: familial hyperchylomicronemia syndrome, F: female, HDL: high-density lipoprotein, LDL: low-density lipoprotein, MCS: multifactorial chylomicronemia, M: male, SBP: systolic blood pressure, TC: total cholesterol, TG: triglycerides, y: years. ^a^ U-Mann–Whitney test, ^b^ Student’s *t*, *p*-values obtained between the groups: ^c^ FCS vs. MCS Children, ^d^ FCS vs. MCS adults, and ^e^ MCS children vs. MCS adults. ^f^ Median [range].

**Table 2 ijms-24-00465-t002:** Genotypic and allelic frequencies of the *APOA5, GPIHBP1,* and *LPL* genes variants detected in patients with primary hypertriglyceridemia.

Location	Identifier	Nucleotide	Amino Acid	Genotypes (*n* = 58)						Alleles (*n* = 116)	
				W/W		W/M		M/M		MA	
				*N*	%	*n*	%	*n*	%	*n*	%
*APOA5*											
5′UTR	rs651821	c.-3A > G		24	41.4	30	51.7	4	6.9	38	32.8
E2	rs3135506	c.56C > G	p.S19W	29	50	23	39.7	6	10.3	35	30.2
E2	rs34282181	c.111C > A	p.D37E	57	98.3	1	1.7	-	-	1	0.9
E2	rs12287066	c.132C > A	p.I44=	27	46.6	25	43.1	6	10.3	37	31.9
I2	rs2072560	c.162-43A > G		3	5.2	24	41.4	31	53.4	86	74.1
E3		c.360G > A ^a^	p.W120*	57	98.3	1	1.7	-	-	1	0.9
E3	rs3135507	c.457G > A	p.V153M	54	93.1	4	6.9	-	-	4	3.6
E3	rs2075291	c.553G > T	p.G185C	56	96.6	2	3.4	-	-	2	1.8
3′UTR	rs619054	c.*31C > T		50	86.2	8	13.8	-	-	8	7.2
3′UTR	rs34089864	c.*76C > T		56	96.6	2	3.4	-	-	2	1.8
*GPIHBP1*											
E1	rs61747644	c.12C > T	p.L4=	29	50	26	44.8	3	5.2	32	27.6
E1	rs11538389	c.41G > T	p.C14F	36	62.1	21	36.2	1	1.7	23	19.8
E1	rs779309481	c.46C > T	p.R16W	56	96.6	2	3.4	-	-	2	1.7
I1	rs1447543669	c.53-2A > G	p.(G18_R60del)	57	98.3	1	1.7	-	-	1	0.9
E2	rs11538388	c.138G > T	p.V46=	25	43.1	28	48.3	5	8.6	33	28.4
I2	rs369934389	c.182-17G > A		53	91.4	5	8.6	-	-	5	4.3
I2	rs773126617	c.182-14C > G		57	98.3	1	1.7	-	-	1	0.9
E3	rs142959160	c.294C > T	p.T98=	57	98.3	1	1.7	-	-	1	0.9
I3	rs56046179	c.295 + 27T > C		6	10.3	15	25.9	37	63.8	89	76.7
I3	rs375588884	c.296-19T > A		57	98.3	1	1.7	-	-	1	0.9
I3	rs141874363	c.295 + 52del		56	96.6	2	3.4	-	-	2	1.7
E4	rs759883512	c.433C > T	p.R145*	57	98.3	1	1.7	-	-	1	0.9
E4	rs751092284	c.460_461del	p.A154Rfs*152	57	98.3	1	1.7	-	-	1	0.9
E4	rs756773250	c.462_463insAAA	p.A154_G155insK	57	98.3	1	1.7	-	-	1	0.9
E4	rs373297994	c.484G > A	p.E162K	54	93.1	4	6.9	-	-	4	3.4
3′UTR	rs1465035770	c.*26C > T		57	98.3	0	0	1	1.7	2	1.7
*LPL*											
E2	rs1801177	c.106G > A	p.D36N	57	98.3	1	1.7	-	-	1	0.9
E2	rs11542065	c.213C > G	p.H71Q	57	98.3	1	1.7	-	-	1	0.9
E3	rs1121923	c.405G > A	p.V135=	57	98.3	1	1.7	-	-	1	0.9
I3	rs343	c.430-34C > A		45	77.6	12	20.7	1	1.7	14	12.1
I3	rs11570897	c.430-6C > T		52	89.7	5	8.6	1	1.7	7	6.0
E4	rs248	c.435G > A	p.E145=	57	98.3	1	1.7	-	-	1	0.9
E5	rs118204057	c.644G > A	p.G215E	57	98.3	1	1.7	-	-	1	0.9
E5		c.723delA ^a^	p.G242Dfs*10	57	98.3	1	1.7	-	-	1	0.9
I5	rs254	c.775 + 33C > G		43	74.1	13	22.4	2	3.4	17	14.7
I5	rs255	c.775 + 37T > C		43	74.1	13	22.4	2	3.4	17	14.7
E6	rs268	c.953A > G	p.N318S	56	96.5	2	3.5	-	-	2	1.8
E8	rs316	c.1164C > A	p.T388=	49	84.5	7	12.1	2	3.4	11	9.5
E8	rs757705770	c.1230G > C	p.K410N	57	98.3	1	1.7	-	-	1	0.9
E9	rs328	c.1421C > G	p.S474*	55	94.8	3	5.2	-	-	3	2.6

E: exon, I: intron, W: wild type, M: mutated; MA: mutated alleles. ^a^ Previously unreported variant.

**Table 3 ijms-24-00465-t003:** Genotypic and allelic frequencies of variants in the *LMF1* gene detected in patients with primary hypertriglyceridemia.

Location	Identifier	Nucleotide	Amino Acid	Genotypes (*n* = 58)						Alleles (*n* = 116)	
				W/W		W/M		M/M		MA	
				*n*	%	*n*	%	*n*	%	*n*	%
P	---	c.-40_-22del ^a^		25	43.1	24	41.4	9	15.5	42	36.2
P	rs13334376	c.-21G > T		45	77.6	12	20.7	1	1.7	14	12.1
E1	rs371796784	c.81T > C	p.P27=	57	98.3	1	1.7	-	-	1	0.9
E1	rs111980103	c.107G > A	p.G36D	49	84.5	9	15.5	-	-	9	7.8
I1	rs113750251	c.194-91G > A		56	96.6	2	3.4	-	-	2	1.7
I1	rs3751665	c.194-77G > A		22	37.9	22	37.9	14	24.2	50	43.1
I1	rs778473079	c.194-64C > T		57	98.3	1	1.7	-	-	1	0.9
I1	rs3751666	c.194-28T > C		20	34.5	15	25.9	23	39.6	61	52.6
E2	rs12448005	c.255T > C	p.L85=	56	96.6	2	3.4	-	-	2	1.7
E2	rs3751667	c.306G > A	p.T102=	30	51.7	21	36.2	7	12.1	35	30.2
E2	rs35663121	c.491T > C	p.V164A	57	98.3	1	1.7	-	-	1	0.9
E4	rs2277892	c.540G > A	p.T180=	37	63.8	18	31	3	5.2	24	20.7
E4	rs2277893	c.543G > A	p.G181=	21	36.2	28	48.3	9	15.5	46	39.7
I4	rs4984706	c.664-58G > C		19	32.8	27	46.5	12	20.7	51	44
I4	rs4984705	c.664-35T > C		19	32.8	27	46.5	12	20.7	51	44
I5	rs751225707	c.729 + 12G > A		57	98.3	1	1.7	-	-	1	0.9
I5	rs11864203	c.729 + 18C > G		12	20.7	26	44.8	20	34.5	66	56.9
I5	rs746716054	c.730-33C > T		57	98.3	1	1.7	-	-	1	0.9
E6	rs2076425	c.756G > A	p.A252=	49	84.5	7	12.1	2	3.4	11	9.5
E6	rs61745065	c.837C > A	p.F279L	57	98.3	1	1.7	-	-	1	0.9
I6	rs113445575	c.898-86C > T		57	98.3	1	1.7	-	-	1	0.9
I6	rs531327980	c.898-85C > T		57	98.3	1	1.7	-	-	1	0.9
E7	rs192520307	c.1052G > A	p.R351Q	57	98.3	1	1.7	-	-	1	0.9
E8	rs35168378	c.1091G > A	p.R364Q	56	96.6	2	3.4	-	-	2	1.7
E8	rs200940009	c.1137C > T	p.S379=	57	98.3	1	1.7	-	-	1	0.9
E8	rs199713950	c.1228G > A	p.G410R	57	98.3	1	1.7	-	-	1	0.9
I8	rs13329717	c.1232 + 54C > G		56	96.6	2	3.4	-	-	2	1.7
I8	rs575396266	c.1233-66C > T		57	98.3	1	1.7	-	-	1	0.9
E9	rs181731943	c.1405G > A	p.A469T	57	98.3	1	1.7	-	-	1	0.9
I9	rs138911395	c.1416 + 24G > A		57	98.3	1	1.7	-	-	1	0.9
E10	rs142258761	c.1476C > T	p.A492=	57	98.3	1	1.7	-	-	1	0.9
E11	rs750316655	c.1638G > A	p.P546=	57	98.3	1	1.7	-	-	1	0.9
E11	rs4984948	c.1685C > G	p.P562R	39	67.2	19	32.8	-	-	19	16.4
3′UTR	rs556385549	c.*27C > T		57	98.3	1	1.7	-	-	1	0.9

P: promoter, E: exon, I: intron, W: wild type, M: mutated; MA: mutated alleles. ^a^ Previously unreported variant.

## Data Availability

Not applicable.

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
