# Peer review of "Biochemical, Clinical, and Genetic Characteristics of Mexican Patients with Primary Hypertriglyceridemia, Including the First Case of Hyperchylomicronemia Syndrome Due to GPIHBP1 Deficiency"

_ijms, 2022, doi:10.3390/ijms24010465_

Round 1
Reviewer 1 Report
The article of Rodríguez-Gutiérrez et al. deals with the description of the clinical and genetic characteristics of a cohort of 58 Mexican patients with chylomicronemia, with the discovery of 2 novel mutations.
The study is interesting and the cohort well described. Nevertheless, several major concerns can be noticed :
1/ the characterization of MCM vs FCS is not clear. As the FCS score (Moulin et al. atherosclerosis 2018 has been used ? Was the classification modified after the results of genetic studies ?Both MCM and FCS definition are unusual.
Chylomicronemia syndrome is defined by TG > 10 mmol/l (or 8.8 g/l) not 4 g/l as used fo MCM in th article. Parents are not needed to have moderate HTG in FCS as heterozygous parents can have normal TG.
It i unusual to has MCM in children. Coud the clinical details /genetic results could be more comment ?
Please clarify.
2/ the discussion is an enumeration of variants identified with description of patients and in silico caracterisation that must be a part of results.
The discussion should be more synthetized for example by genes, and focused on results comment and on the two new variants identified.
3/ Why APOC2 gene was not sequenced ? do all patients had a apoC2 dosage to exclude apoC2 deficiencies ? APOE variants frequently associated with are also not given
4/ The genetic studies were done by direct sequencing and not by NGS which does not allow to detect copy number variation. This should be comment and added as a limit of the study
5/ Do some patient have no variants ? for frequent variants, the frequency could be compared in the table to general population frequency such as in 1000 genome database.
Minor concern :
Some reference are incomplete to verify.
i.e n° 12 some authors are lacking at the end.
Serveaux Dancer M, Di Filippo M, Marmontel O, Valéro R, Piombo Rivarola MDC, Peretti N, Caussy C, Krempf M, Vergès B, Mahl M, Marçais C, Moulin P, Charrière S.
Author Response
Biochemical, clinical, and genetic characteristics of Mexican patients with primary hypertriglyceridemia, including the first case of hyperchylomicronemia syndrome due to GPIHBP1 deficiency
I thank all reviewers for their comments and suggestions to improve this manuscript.
RESPONSE TO REVIEWER 1
The article of Rodríguez-Gutiérrez et al. deals with the description of the clinical and genetic characteristics of a cohort of 58 Mexican patients with chylomicronemia, with the discovery of 2 novel mutations.
The study is interesting and the cohort well described. Nevertheless, several major concerns can be noticed :
1) The characterization of MCM vs FCS is not clear. As the FCS score (Moulin et al. atherosclerosis 2018 has been used ? Was the classification modified after the results of genetic studies ?Both MCM and FCS definition are unusual.
ANSWER: The diagnostic criteria paragraph was modified in the document; in addition, both MCM and FCS definition was changed according the bibliography proposed by the reviewer was considered (Moulin et al. 2018).
2) Chylomicronemia syndrome is defined by TG > 10 mmol/l (or 8.8 g/l) not 4 g/l as used fo MCM in th article. Parents are not needed to have moderate HTG in FCS as heterozygous parents can have normal TG.
ANSWER: The manuscript was modified.
3) It is unusual to has MCM in children. Could the clinical details /genetic results could be more comment?
ANSWER: In the last decade, the cases of children with dyslipidemia have increased in Mexico. We discuss only those index cases that showed some confirmed pathogenic variant. However, Supplementary table 1 all the information of each patient studied is placed.
Please clarify.
4) the discussion is an enumeration of variants identified with description of patients and in silico caracterisation that must be a part of results.
ANSWER: We structured the results and discussion sections.
5) The discussion should be more synthetized for example by genes, and focused on results comment and on the two new variants identified.
ANSWER: We describe each variant that has been described as pathogenic and the clinical characteristics of the index case that was heterozygous. In addition, we mention the common variants that have been associated with HTG; this in an attempt to integrate the possible genetic causes of the disease. We were as specific as possible. We did many drafts to present the information and it was the one that seemed clearest and least tedious to us.
6) Why APOC2 gene was not sequenced? do all patients had a apoC2 dosage to exclude apoC2 deficiencies ? APOE variants frequently associated with are also not given.
ANSWER: Financial resources were scarce. In my country there were administrative changes and, in 2020, the financing we had was withdrawn, so we screened only 4 genes. To date we have not had access to financing. Regarding the APOE isoforms, we had the opportunity to analyze 24 patients and 18 (75%) were E3/E3, 5 (20.8%) E3/E4, and 1 (4.1%) E2/E3. The remaining 34 patients were not screened because the primers were finished and considering the frequencies obtained in the 24 patients, we believed that it was not very significant.
7) The genetic studies were done by direct sequencing and not by NGS which does not allow to detect copy number variation. This should be comment and added as a limit of the study
ANSWER: We added this point in conclusions.
8) Do some patient have no variants ? for frequent variants, the frequency could be compared in the table to general population frequency such as in 1000 genome database.
ANSWER: All patients had variants, each patient had a minimum of eight and a maximum of 22 variants of the four screened genes (Supplementary table). This information is described in the manuscript. On the other hand, the common variants were compared with the percentages described in 1000 genome database; this comparison showed the association of four variants (rs651821, rs3135506, and rs3751666) with the HTG trait.
Minor concern :
Some reference are incomplete to verify.
i.e n° 12 some authors are lacking at the end.
Serveaux Dancer M, Di Filippo M, Marmontel O, Valéro R, Piombo Rivarola MDC, Peretti N, Caussy C, Krempf M, Vergès B, Mahl M, Marçais C, Moulin P, Charrière S.
ANSWER: References were corrected.
Reviewer 2 Report
The authors aimed to describe biochemical-clinical characteristics and variants in the APOA5, GPIHBP1, LMF1, and LPL genes in 58 unrelated Mexican patients with severe primary HTG. Thirty DNA fragments were analyzed using PCR and Sanger sequencing. Seventy-four variants were found (10 in APOA5, 16 in GPIHBP1, 34 in LMF1, and 14 in LPL), of which 15 could be involved in the development of primary HTG. They report the first Mexican patient with hyperchylomicronemia syndrome due to GPIHBP1 deficiency caused by three variants. Moreover, twelve patients were heterozygous for the rare variants described as causing primary HTG and also presented common variants of risk, which could partially explain their phenotype. As finding, two novel LMF1 and LPL genetic variants were identified.
Genetic mapping of cases with familial chylomicronemia syndrome is a crucial step to diagnose and treat these severe cases. Therefore, any novel information about the genetic background of these patients can be important for clinicians. Although the study reports some interesting novel genetic variants that may involved in the development of hyperchylomicronemia, the diagnostic process used for the identification of FCS and MCS patients should be clarified.
Comments:
1. The nomenclature should be corrected: multifactorial chylomicronemia syndrome (MCS) should be used instead of multifactorial chylomicronemia (MCM).
2. The diagnostic algorithm used for the diagnosis of FCS and MCS is unclear. Determination of LPL activity and/or lipoprotein elecrophoresis should be used to prove the diagnosis of FCS.
3. Alternatively, the FCS score developed by Moulin et al. could be used as an easy-to-use diagnostic tool for the identification of FCS patients.
4. Data on classical clinical characteristics of FCS including eruptive xanthoma, lipemia retinalis, recurrent abdominal pain, and hepatosplenomegaly should be added. Some other parameters, such as blood pressure values are not necessary. Indeed, because of the low number of FCS cases, statistical comparison of FCS and MCS patients’ data might be problematic.
5. To prove the causal role of the reported GPIHBP1 variants in FCS measurement of significantly decreased LPL activity are essential.
6. Data on medication should be mentioned.
7. English needs editing.
Author Response
Biochemical, clinical, and genetic characteristics of Mexican patients with primary hypertriglyceridemia, including the first case of hyperchylomicronemia syndrome due to GPIHBP1 deficiency
I thank all reviewers for their comments and suggestions to improve this manuscript.
RESPONSE TO REVIEWER 2
The authors aimed to describe biochemical-clinical characteristics and variants in the APOA5, GPIHBP1, LMF1, and LPL genes in 58 unrelated Mexican patients with severe primary HTG. Thirty DNA fragments were analyzed using PCR and Sanger sequencing. Seventy-four variants were found (10 in APOA5, 16 in GPIHBP1, 34 in LMF1, and 14 in LPL), of which 15 could be involved in the development of primary HTG. They report the first Mexican patient with hyperchylomicronemia syndrome due to GPIHBP1 deficiency caused by three variants. Moreover, twelve patients were heterozygous for the rare variants described as causing primary HTG and also presented common variants of risk, which could partially explain their phenotype. As finding, two novel LMF1 and LPL genetic variants were identified.
Genetic mapping of cases with familial chylomicronemia syndrome is a crucial step to diagnose and treat these severe cases. Therefore, any novel information about the genetic background of these patients can be important for clinicians. Although the study reports some interesting novel genetic variants that may involved in the development of hyperchylomicronemia, the diagnostic process used for the identification of FCS and MCS patients should be clarified.
Comments:
- The nomenclature should be corrected: multifactorial chylomicronemia syndrome (MCS) should be used instead of multifactorial chylomicronemia (MCM).
ANSWER: The manuscript was modified.
- The diagnostic algorithm used for the diagnosis of FCS and MCS is unclear. Determination of LPL activity and/or lipoprotein elecrophoresis should be used to prove the diagnosis of FCS. Alternatively, the FCS score developed by Moulin et al. could be used as an easy-to-use diagnostic tool for the identification of FCS patients.
ANSWER: The diagnostic criteria paragraph was modified in the document. We used some diagnostic criteria proposed by Moulin et al. 2018. We were unable to perform the electrophoresis, but several criteria defined by Moulin et al. were observed in our patients, so we believe that the diagnosis is well made. Unfortunately, in three patients with FCS no homozygous or compound heterozygous genetic variants were found, but we cannot rule out the clinical and biochemical diagnosis.
- Data on classical clinical characteristics of FCS including eruptive xanthoma, lipemia retinalis, recurrent abdominal pain, and hepatosplenomegaly should be added. Some other parameters, such as blood pressure values are not necessary. Indeed, because of the low number of FCS cases, statistical comparison of FCS and MCS patients’ data might be problematic.
ANSWER: None of our patients presented retinal xanthomas or lipemia (this information is described in the manuscript); and hepatosplenomegaly was not evaluated by the treating physician. Moulin et al. mention: "Other clinical symptoms include transient eruptive xanthomas, which affect <50% of individuals with FCS, often appearing on the trunk and extremities, and lipemia retinalis, a milky appearance of the retinal vessels”. However, we previously reported a case with LPL deficiency who did present xanthomas from the age of 4 (perhaps due to the more severe phenotype) (Colima Fausto, A.G.et al. Ann. Lab. Med. 2017, 37, 355–358).
On the other hand, I consider that it is important to report concomitant diseases that patients present, since it is more complicated to control TG levels. There is even a record that some patients developed type 2 DM after episodes of pancreatitis.
- To prove the causal role of the reported GPIHBP1 variants in FCS measurement of significantly decreased LPL activity are essential.
ANSWER: I know it is important to measure LPL activity, but it is impossible for us, since the patient did not return for consultation and he decided not to study his relatives.
- Data on medication should be mentioned.
ANSWER: In most of the patients that we find a pathogenic variant, we mention the treatment. We include a column in the supplementary table with the treatment.
- English needs editing.
ANSWER: The English was revised and some details were changed.
Reviewer 3 Report
The article is of great interest. The results are of scientific novelty. I have no fundamental remarks. The article may be published.
Author Response
Biochemical, clinical, and genetic characteristics of Mexican patients with primary hypertriglyceridemia, including the first case of hyperchylomicronemia syndrome due to GPIHBP1 deficiency
Response to Reviewer 3
The article is of great interest. The results are of scientific novelty. I have no fundamental remarks. The article may be published.
ANSWER: I thank for their comments to this manuscript.
Reviewer 4 Report
Summary: The authors clearly identify the need for this study in the introduction. The report regarding variant frequencies in hypertriglyceridemia in a Mexican population is straightforward. Perhaps the some of the most valuable information is in the discussion, where the findings are discussed relative to those in other populations. In addition, a number of new variants were reported in this population. Clinicians and geneticists interested in hypertriglyceridemia will find this report to be of interest.
Major comments:
1. Lines 78-79: The authors state that HTG is one of the most frequent dyslipidemias in Mexico. Please report its frequency and cite a source for this information. See also line 170, where it is described as “common”.
2. Please move the conclusions to immediately follow the discussion.
Minor comments:
1. Line 26: “whose” should only apply to people. Please change the wording to “…a health problem for which the genetic bases…”
2. Lines 38 and 84: “As result” is not idiomatic English. This should be “As a result”. Alternatively, this phrase could be left out entirely without affecting the meaning.
3. Line 163: in silico analyzes should be analyses. Please be consistent with italicization of “in silico” throughout the manuscript
4. Line 164: “predicted that it as disease-causing”. This is awkward language. Please correct.
5. Line 405: “analyzes” should be “analyses”.
Author Response
Biochemical, clinical, and genetic characteristics of Mexican patients with primary hypertriglyceridemia, including the first case of hyperchylomicronemia syndrome due to GPIHBP1 deficiency
I thank all reviewers for their comments and suggestions to improve this manuscript.
RESPONSE REVIEWER 4
Summary: The authors clearly identify the need for this study in the introduction. The report regarding variant frequencies in hypertriglyceridemia in a Mexican population is straightforward. Perhaps the some of the most valuable information is in the discussion, where the findings are discussed relative to those in other populations. In addition, a number of new variants were reported in this population. Clinicians and geneticists interested in hypertriglyceridemia will find this report to be of interest.
Major comments:
1) Lines 78-79: The authors state that HTG is one of the most frequent dyslipidemias in Mexico. Please report its frequency and cite a source for this information. See also line 170, where it is described as “common”.
ANSWER: In Mexico, there have been some reports on the frequency of hypertiglyceridemia derived from population screening. However, due to the number of references required by the journal, we only mention the percentages specifically observed from the region from which the patients were recruited.
2) Please move the conclusions to immediately follow the discussion.
ANSWER: The conclusions were placed in the order requested by the journal.
Minor comments:
3) Line 26: “whose” should only apply to people. Please change the wording to “…a health problem for which the genetic bases…”
ANSWER: The manuscript was modified.
4) Lines 38 and 84: “As result” is not idiomatic English. This should be “As a result”. Alternatively, this phrase could be left out entirely without affecting the meaning.
ANSWER: The manuscript was modified.
5) Line 163: in silico analyzes should be analyses. Please be consistent with italicization of “in silico” throughout the manuscript.
ANSWER: The manuscript was modified.
6) Line 164: “predicted that it as disease-causing”. This is awkward language. Please correct.
ANSWER: The manuscript was modified.
7) Line 405: “analyzes” should be “analyses”.
ANSWER: The manuscript was modified.
Round 2
Reviewer 1 Report
The manuscript was improved to answer to reviewer comments.
Nevertheless the main point about the structure of results and discussion with a list of variants was not modified significantly. It was only fusionned in one part number 2 with results en discussion. Paragraph number 3 is missing. to correct.
Author Response
Reviewer 1
- Nevertheless, the main point about the structure of results and discussion with a list of variants was not modified significantly.
ANSWER: Although his suggestion of structuring the discussion according to genes and not variants was not very clear to us, we made various attempts to determine the best way to present the results and discuss them. We consider presenting the results and discussion sections together as the best option. The patient with hyperchylomicronemia syndrome is presented first due to variants in the GPIHBP1 gene. Variants of each gene are placed in alphabetical order (APOA5, GPIHBP1, LMF1, and LPL genes). In order to explain the phenotype of hypertriglyceridemia, we commented the relevant characteristics of each patient presenting a “pathogenic variant”, as well as, those polymorphisms that resulted with significant Odds.
- It was only fusionned in one part number 2 with results en discussion. Paragraph number 3 is missing. to correct.
ANSWER. The manuscript was modified.
Reviewer 2 Report
Although some of the suggested corrections have been made, several clinical data and laboratory parameters are missing. There are basic methodological problems in the diagnostic process. Indeed, the causal role of the novel mutation is still questionable. Therefore, the manuscript cannot be accepted.
Author Response
Reviewer 2
- Although some of the suggested corrections have been made, several clinical data and laboratory parameters are missing.
ANSWER: Unfortunately, it was not possible to obtain all clinical and biochemical data of all patients, but we consider that the essential data to establish the diagnosis of primary hypertriglyceridemia are complete.
- There are basic methodological problems in the diagnostic process.
ANSWER: We consider that the diagnosis of primary hypertriglyceridemia is right in our cases. All patients had a documented history of recurrent HTG, high triglyceride levels at diagnosis, and most important, we detected first-degree relatives with the biochemical trait of HTG; together, these features are sufficient to diagnose primary hypertriglyceridemia. Patients were classified as FCS taking into account TG levels (>885 mg/dL) from the first decade of life and without secondary factors, a TG/TC ratio >5, epigastric pain, hypertriglyceridemia-induced acute pancreatitis, and homozygous or composite heterozygous coding variant APOA5, GPIHBP1, LPL, or LMF1. The supplementary table highlights the 4 patients with hyperchylomicronemia syndrome and, three of them (54, 56 and 57), met all the characteristics described by Moullin et al. (2018). Unfortunately, the fourth patient no longer came to the clinic for care, so we were unable to obtain some clinical data, however, he presented all three variants in the GPIHBP1 gene; in addition, he began with HTG problems since the first decade of life.
On the other hand, patients with MCS had more than one of the following characteristics: moderate TG levels (>200 mg/dL), secondary factors triggering HTG (high alcohol intake, features of metabolic syndrome, overweight), TG reduction with treatment and diet, as well as pancreatitis or epigastric pain. Therefore, it is not clear to us the reviewer’s comment underestimating a work that took us several years to biochemically analyze so many families and later, select those that had primary HTG.
3) Indeed, the causal role of the novel mutation is still questionable.
ANSWER: Two mutations were novel: c.-40_-22del LMF1 and p.G242Dfs*10 LPL. Actually, the first one (c.-40_-22del LMF1) can be considered as benign, since family analysis showed that normotriglyceridemic patients were homozygous for the variant, therefore, it is not the cause of HTG; although in vivo studies could be convenient to confirm this. The second (p.G242Dfs*10 LPL) is a novel variant which resulted from an adenine deletion at nucleotide 723. This deletion produces a truncated protein of 250 amino acids that lacks 225 residues, resulting in the loss of both a segment of the amino-terminal domain and the entire carboxyl terminal domain. These regions are essential for the binding of LPL protein to lipid and heparin molecules, as well as to the GPIHBP1 and LRP1 proteins [36,38,40,41]. We did not perform in vivo or in vitro assays, but we consider that it is pathogenic since more than 50% of the protein is lost and the eliminated domains are essential for LPL dimerization, which is the active form of the protein.
- Therefore, the manuscript cannot be accepted.
ANSWER: We respect your opinion but we disagree.
Our work provides a lot of new and valuable information in the field of HTG, which is an area little studied in the Mexican population and represents a serious public health problem.
Round 3
Reviewer 2 Report
I accept the answers.